# Institutional differences in USMLE Step 1 and 2 CK performance: Cross-sectional study of 89 US allopathic medical schools

**Jesse Burk-Rafel** [1]*, **Ricardo W. Pulido**[2], **Yousef Elfanagely**[3], **Joseph C. Kolars**[4]

**1** Department of Internal Medicine, New York University Langone Health, New York, NY, United States of America, **2** Department of Otolaryngology–Head and Neck Surgery, University of Washington, Seattle, WA, United States of America, **3** Department of Internal Medicine, Brown University, Providence, RI, United States of America, **4** Department of Internal Medicine, University of Michigan Medical School, Ann Arbor, MI, United States of America

* jesse.rafel@nyulangone.org

## Abstract

### Introduction

The United States Medical Licensing Examination (USMLE) Step 1 and Step 2 Clinical Knowledge (CK) are important for trainee medical knowledge assessment and licensure, medical school program assessment, and residency program applicant screening. Little is known about how USMLE performance varies between institutions. This observational study attempts to identify institutions with above-predicted USMLE performance, which may indicate educational programs successful at promoting students' medical knowledge.

### Methods

Self-reported institution-level data was tabulated from publicly available *US News and World Report* and Association of American Medical Colleges publications for 131 US allopathic medical schools from 2012–2014. Bivariate and multiple linear regression were performed. The primary outcome was institutional mean USMLE Step 1 and Step 2 CK scores outside a 95% prediction interval ($\geq 2$ standard deviations above or below predicted) based on multiple regression accounting for students' prior academic performance.

### Results

Eighty-nine US medical schools (54 public, 35 private) reported complete USMLE scores over the three-year study period, representing over 39,000 examinees. Institutional mean grade point average (GPA) and Medical College Admission Test score (MCAT) achieved an adjusted $R^2$ of 72% for Step 1 (standardized $\beta_{MCAT}$ 0.7, $\beta_{GPA}$ 0.2) and 41% for Step 2 CK (standardized $\beta_{MCAT}$ 0.5, $\beta_{GPA}$ 0.3) in multiple regression. Using this regression model, 5 institutions were identified with above-predicted institutional USMLE performance, while 3 institutions had below-predicted performance.

**Data Availability Statement:** All relevant data are within the paper and its Supporting Information files.

**Funding:** The authors received no specific funding for this work.

**Competing interests:** We have read the journal's policy and the authors of this manuscript have the following competing interests: Dr. Burk-Rafel reports working as a research consultant for ScholarRx, a digital learning platform that includes USMLE preparation services, during the late stages of writing this manuscript. ScholarRx was not involved in this study in any way. All other authors declare no competing interests. This does not alter our adherence to PLOS ONE policies on sharing data and materials.

## Conclusions

This exploratory study identified several US allopathic medical schools with significant above- or below-predicted USMLE performance. Although limited by self-reported data, the findings raise questions about inter-institutional USMLE performance parity, and thus, educational parity. Additional work is needed to determine the etiology and robustness of the observed performance differences.

## Introduction

The United States Medical Licensing Examination (USMLE) is a 3-step examination required for medical licensure in the United States. The first two exams, USMLE Step 1 and Step 2 Clinical Knowledge (CK), assess medical students' mastery of basic biomedical principles and their clinical applications [1,2]. About 40,000 trainees take each exam annually, of which over 35% are non-US/Canadian medical students [3]. Both exams are high-stakes parameters of medical student performance critical for advancement [4], residency applicant screening and selection [5,6], and future board certification [7]. Multiple studies have demonstrated correlations between *individual* factors–including Medical College Admission Test (MCAT) score [8], undergraduate grade point average (GPA) [9], and study behaviors [10]–and USMLE performance. However, little is known about *institutional* USMLE performance variation. One group analyzing data from the 1990s demonstrated that institutional variables, including curricular differences, did not predict USMLE performance [11,12]. A recent study using one year of national data found some evidence of inter-institutional USMLE performance differences, but the short study duration precludes definitive conclusions [13].

In this exploratory, institution-level study, we analyze institutional variation in USMLE Step 1 and Step 2 CK performance relative to mean matriculant GPA and MCAT. Our primary objective was to identify institutions with above-predicted USMLE performance, which may indicate educational programs successful at promoting students' medical knowledge.

## Methods

This observational study was conducted in accordance with the STROBE guidelines for observational studies in epidemiology [14].

### Data sources

We manually tabulated self-reported institutional data–aggregate percentages and means representing yearly medical student cohorts at single institutions–from the annual *US News and World Report* "Best Graduate Schools" publication (2008–2016 editions) [15] and the Association of American Medical Colleges (AAMC) *Medical School Admission Requirements* publication (2008–2012 editions) [16] for all 131 US allopathic medical schools. Osteopathic institutions were excluded from this study, as osteopathic students typically take the COMLEX licensing examination rather than the USMLE and very few US osteopathic institutions reported USMLE performance data. A sample size calculation was not performed because we obtained available data for a census of US allopathic medical schools during the study period. National averages for all allopathic matriculants and examinees were obtained from official AAMC [17] and USMLE sources [18,19]. Institutional Review Board approval was not required as no human subjects or identifiable data were involved.

### Primary outcome measures and predictor variables

The primary outcome measures were institutional mean USMLE Step 1 and 2 CK scores, averaged over the 3-year study period 2012–2014. Predictor variables included *students' prior academic performance* (institutional mean undergraduate GPA and MCAT, averaged over 3 years) and *demographics* (percentage non-traditional students, minority students, undergraduate biological sciences or humanities majors), and *medical school factors* (acceptance rate, public/private status, faculty-to-student ratio, National Institutes of Health research funding, graduates entering primary care). MCAT scores represented total scores computed as the sum of the average institutional scores on all 3 sections (biological sciences, physical sciences, verbal reasoning). Institutional USMLE scores were matched to institutional GPA and MCAT averages from two or four years prior (for Step 1 or 2 CK, respectively) to account for the typical lag between matriculation and USMLE testing.

### Statistical analysis

All analysis was at the institution level. We performed ordinary least squares linear regression analysis, with test of Pearson's *r* for bivariate correlations. Conditions of linearity, nearly normal residuals, and homoscedasticity were checked [20]. Institutions with 3-year average USMLE performance outside a 95% prediction interval (regression residual ≥2 standard deviations, SD, from predicted) were identified [21]. Hypothesis tests were 2-sided with α = .05; ANOVA was used to confirm overall significance of multiple regressions. Statistical analysis was done using SPSS version 25.0 (SPSS Inc., Chicago, Illinois).

## Results

In total, 89 (54 public and 35 private) of 131 US allopathic medical schools reported complete USMLE scores over the 3-year study period (68% reporting rate), representing 39,615 and 39,252 Step 1 and 2 CK examinees, respectively. Among reporting institutions, the institutional mean USMLE Step 1 score was 229.7 (SD 5.5) and Step 2 CK score was 238.3 (SD 4.7) (Table 1). GPA and MCAT scores showed minimal heterogeneity across the study years (data not shown). USMLE scores increased across the study years, which was also observed nationally. The average GPA, MCAT scores, and USMLE Step 1 scores for the 89 reporting institutions were slightly higher than national averages for all matriculants/examinees. Complete GPA, MCAT, and USMLE data for reporting institutions and nationally are provided in S1 Table.

### Predictors of institutional USMLE performance

The strongest predictor of institutional USMLE scores was prior student academic performance, including undergraduate GPA (Step 1, Pearson's *r* = .64; Step 2 CK, *r* = .53; both *P* < .001) and MCAT score (Step 1, *r* = .84; Step 2 CK, *r* = .62; both *P* < .001). Numerous student body demographic and institutional factors had moderately strong correlations with institutional USMLE scores in bivariate regression; however, when controlling for GPA and MCAT, these correlations were weak and no longer significant (Table 2). For example, private institutions were correlated with higher USMLE Step 1 scores (*r* = .51, *P* < .001), but this correlation vanished after controlling for GPA and MCAT (*r* = .12, *P* = .42), as private institutions recruit students with higher MCAT scores compared to public institutions (mean 33.5 vs. 30.9, difference 2.7, 95% CI 1.9–3.5; *P* < .001).

The final regression model utilizing GPA and MCAT achieved an adjusted $R^2$ of 72% for Step 1 (standardized $\beta_{MCAT}$ 0.7, $\beta_{GPA}$ 0.2, model *P* < .001) and 41% for Step 2 CK

**Table 1. Average GPA, MCAT, and USMLE Step 1 and 2 CK score among 89 US allopathic medical schools and nationally.**

| | All Schools (n = 89) Average (SD) | Public (n = 54) Average (SD) | Private (n = 35) Average (SD) | P value* | National Average |
|---|---|---|---|---|---|
| **GPA 2010–2012** | 3.70 (0.08) | 3.68 (0.07) | 3.73 (0.08) | ns | 3.67 |
| **MCAT 2010–2012** | 31.9 (2.2) | 30.9 (1.7) | 33.5 (2.0) | < .001 | 31.1 |
| **USMLE Step 1** | | | | | |
| 2012 | 227.6 (6.1) | 225.4 (5.3) | 230.9 (5.6) | < .001 | 227 |
| 2013 | 230.4 (5.8) | 228.1 (4.9) | 233.9 (5.3) | < .001 | 228 |
| 2014 | 231.1 (5.6) | 229.0 (4.6) | 234.4 (5.6) | < .001 | 229 |
| 2012–2014 (combined) | 229.7 (5.5) | 227.5 (4.4) | 233.1 (5.2) | < .001 | 228.0 |
| **USMLE Step 2 CK** | | | | | |
| 2012 | 235.6 (5.5) | 234.7 (5.1) | 237.0 (5.8) | ns | 237 |
| 2013 | 238.8 (5.3) | 237.6 (4.8) | 240.7 (5.5) | < .01 | 238 |
| 2014 | 240.5 (4.5) | 239.3 (4.1) | 242.4 (4.4) | < .01 | 240 |
| 2012–2014 (combined) | 238.3 (4.7) | 237.2 (4.2) | 240.0 (4.9) | < .01 | 238.3 |

GPA, Undergraduate Grade Point Average; MCAT, Medical College Admission Test score; USMLE, US Medical Licensing Examination; CK, Clinical Knowledge; ns, not significant at $P < .05$ threshold

* Two-tailed t-test comparing public to private

(standardized $\beta_{MCAT}$ 0.5, $\beta_{GPA}$ 0.3, model $P < .001$). GPA added significant but incremental validity evidence over MCAT alone (Step 1, $\Delta R^2$ 2%, $P = .009$; Step 2 CK, $\Delta R^2$ 4%, $P = .02$); accordingly, for visualization, institutional USMLE was regressed on MCAT score alone (Fig 1).

**Table 2. Linear regression between various institutional characteristics and institutional USMLE performance, without and with control for average institutional GPA and MCAT.**

| USMLE Step 1 | Pearson's r | Partial ρ† | USMLE Step 2 CK | Pearson's r | Partial ρ‡ |
|---|---|---|---|---|---|
| Institutional GPA | .64** | – | Institutional GPA | .53** | – |
| Institutional MCAT | .84** | – | Institutional MCAT | .62** | – |
| USMLE Step 2 CK | .56** | .05 | USMLE Step 1 | .56** | .06 |
| Minority Students | .46** | .16 | Minority Students | .25* | -.03 |
| Biological Science Majors | -.36** | -.07 | Biological Science Majors | -.27* | -.12 |
| Humanities Majors | .13 | .07 | Humanities Majors | .10 | .11 |
| Non-Traditional Students | .01 | -.13 | Non-Traditional Students | -.03 | -.07 |
| Acceptance Rate | -.30** | -.14 | Acceptance Rate | -.17 | -.05 |
| Private Institution | .51** | .12 | Private Institution | .30** | -.06 |
| Faculty:Student Ratio | .44** | .01 | Faculty:Student Ratio | .35** | .06 |
| NIH Funding | .58** | -.13 | NIH Funding | .47** | -.01 |
| Primary Care Grads | -.31** | -.12 | Primary Care Grads | -.10 | .17 |

GPA, Undergraduate Grade Point Average; MCAT, Medical College Admission Test score; USMLE, US Medical Licensing Examination; CK, Clinical Knowledge; NIH, National Institutes of Health.

* $P < .05$

** $P < .01$

† Partial correlation controlling for GPA and MCAT (2010–12)

‡ Partial correlation controlling for GPA and MCAT (2008–10)

**(A)**

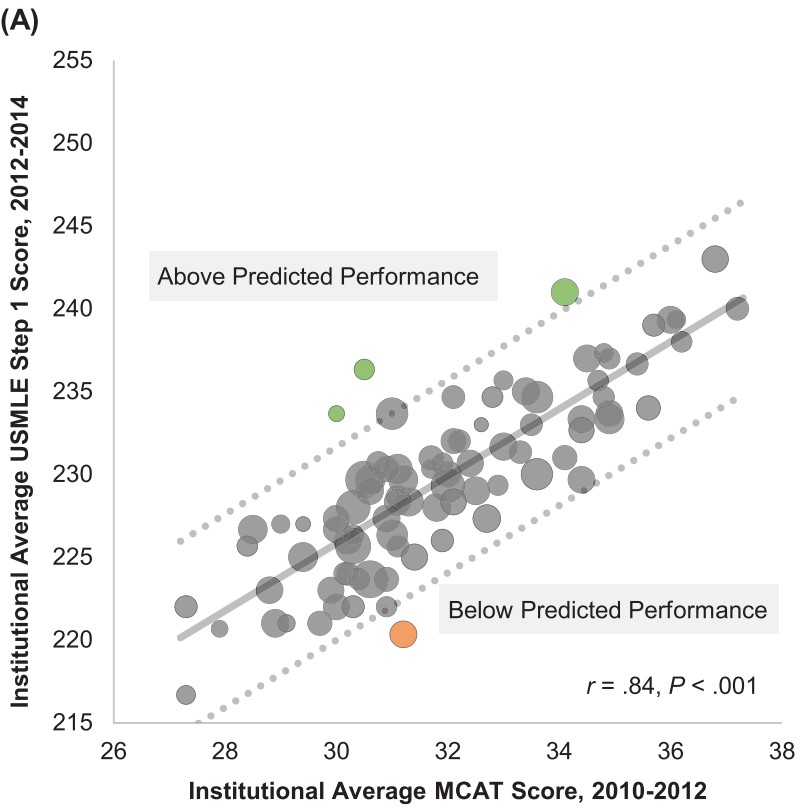

**(B)**

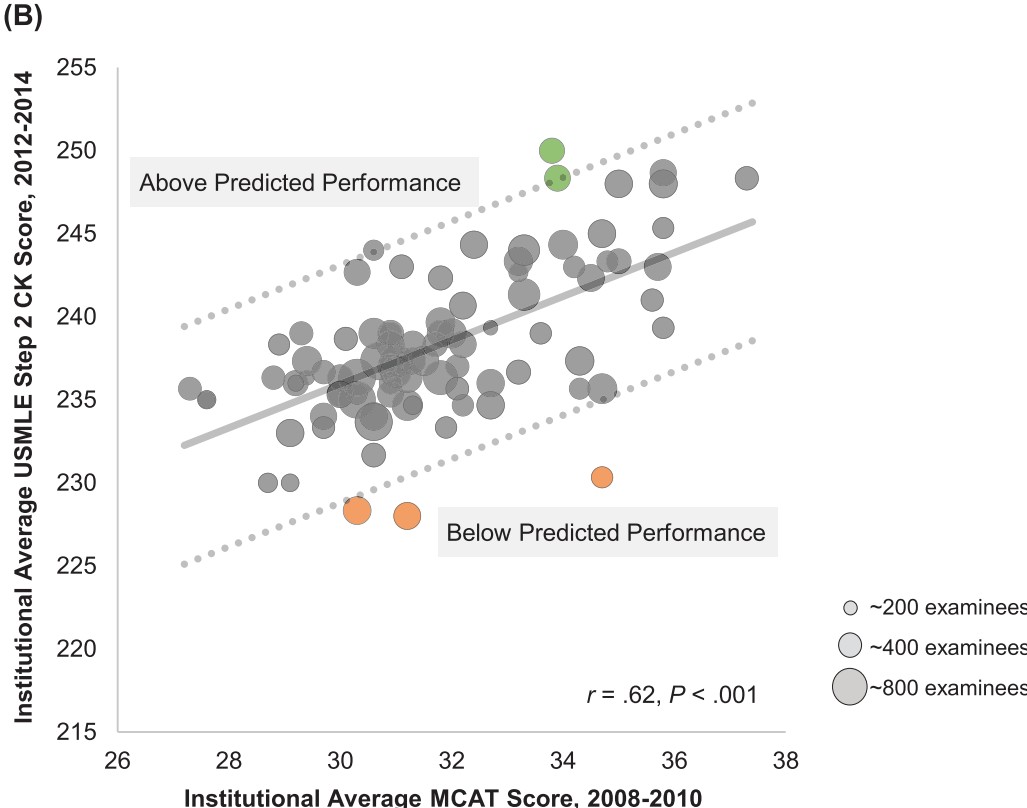

**Fig 1. Regression analysis of institutional MCAT versus USMLE performance.** (A) Regression analysis of institutional average matriculant Medical College Admission Test (MCAT) score (2010–2012) versus institutional average US Medical Licensing Examination (USMLE) Step 1 score (2012–2014) across n = 89 US allopathic medical schools, representing 39,615 examinees. (B) Regression analysis of institutional average matriculant MCAT score (2008–2010) versus institutional average USMLE Step 2 Clinical Knowledge (CK) score (2012–2014) across n = 89 US allopathic medical schools, representing 39,252 examinees. For both plots, each bubble represents 3-year average at one institution, with bubble size reflecting number examined at each institution. Ordinary least squares best fit line (solid) and 95% prediction interval (dashed lines) are shown, with colored data points highlighting institutions outside the prediction interval.

## Institutions with above- or below-predicted USMLE performance

Using the GPA and MCAT regression model, we identified a subset of institutions with 3-year average institutional USMLE scores statistically above or below predicted (Table 3).

## Discussion

In this exploratory study of 89 US allopathic medical schools, we identified 5 institutions with above-predicted institutional USMLE performance based on the described model. The etiology of these institutions' relative success (or the 3 unnamed institutions' below-predicted performance) is unclear; we can only say that numerous demographic and institutional factors we assessed did not account for this variation. We hypothesize that unmeasured student factors that vary systematically between institutions (e.g., through admissions processes) or institution-specific factors (e.g., alignment of curricula with USMLE content) may explain these institutional differences. For example, medical schools that provide commercially available Step 1 question banks [22] or where students take Step 1 after the core clerkships [23] have reported improved institutional scores, demonstrating that unique institutional strategies can promote students' USMLE success. Further study is needed to understand if the 5 institutions identified here have unique factors that promoted their students' success on these exams.

We found that institutions' average student GPA and MCAT accounted for substantial variation in institutional average USMLE Step 1 and Step 2 CK scores, which was expected based on prior studies at the individual [8,9] and institutional level [9,11,12]. Importantly, institutional demographic factors (such as percent minority students or biological sciences majors) were correlated with institutional USMLE performance in bivariate regression but were not significant after controlling for GPA and MCAT. National Institutes of Health research funding, which had been previously shown to correlate with institutional USMLE performance [13], was similarly not significant when controlling for GPA and MCAT.

Institutions with a propensity for matching students in the primary care specialties family medicine, pediatrics, and internal medicine–which have lower USMLE screening thresholds for residency interviews than other more "competitive" specialties [24]–tended to recruit students with lower GPA and MCAT scores, and thus lower institutional USMLE scores. As with other institutional factors, however, institutions' primary care specialty rate was not associated with differential USMLE performance beyond its association with GPA and MCAT.

Such findings highlight the critical importance of controlling for prior academic performance when attempting to explain USMLE performance differences. However, we doubt that pre-medical students–a key consumer of the annual *US News and World Report* data–consider these covariates when interpreting institutional USMLE scores and identifying medical schools of interest. Indeed, undergraduates might conclude (erroneously) that private medical schools outperform public schools on the USMLE, when in fact students attending private schools have higher test scores at matriculation. There may be a role for better contextualizing this data so that pre-medical students can be informed consumers. The National Board of Medical

**Table 3. US allopathic medical schools with above- or below-predicted[a] institutional USMLE Step 1 or Step 2 CK performance, 2012–2014.**

| USMLE Step 1 | | | | |
| --- | --- | --- | --- | --- |
| **Institution** | **Average Score (SD)** | **Score Deviation from Predicted, Points** | **Standardized Residual, SD** | **Examinees, No.** |
| University of Hawaii–Manoa | 234 (3.2) | +8.4 | +2.9 | 182 |
| University of Missouri | 236 (4.6) | +8.2 | +2.8 | 296 |
| Baylor College of Medicine | 241 (1.0) | +5.9 | +2.0 | 517 |
| Institution X[b] | 220 (3.2) | -5.9 | -2.0 | 504 |
| **USMLE Step 2 CK** | | | | |
| **Institution** | **Average Score (SD)** | **Score Deviation from Predicted, Points** | **Standardized Residual, SD** | **Examinees, No.** |
| Emory University | 250 (2.6) | +10.1 | +2.8 | 424 |
| University of Virginia | 248 (2.1) | +7.1 | +2.0 | 449 |
| Institution X | 228 (5.3) | -7.3 | -2.0 | 481 |
| Institution Y | 228 (9.7) | -9.2 | -2.6 | 507 |
| Institution Z | 230 (3.2) | -12.0 | -3.4 | 305 |

USMLE, US Medical Licensing Examination; CK, Clinical Knowledge; SD, standard deviation.

[a] Based on regression models incorporating institutional average Medical College Admission Test (MCAT) score and undergraduate grade point average (GPA) of entering students, as follows: Institutional USMLE Step 1 score = 122 + 1.7 * MCAT + 14.1 * GPA; Institutional USMLE Step 2 CK score = 149 + 1.0 * MCAT + 15.6 * GPA.

[b] The names of institutions with below-predicted institutional USMLE performance were withheld due to the sensitive and exploratory nature of this data.

Examiners (NBME), who produce the USMLE, are positioned to more rigorously explore the relationship between institutions and exam performance.

## Limitations

This study relied on self-reported institutional data via a third-party publication, as the NBME does not publish institutional score performance. Misreporting is possible, although we validated the reported scores from several institutions. *US News and World Report* provides their methodology for data collection with each annual release [25], but do not state specifics related to how data is validated or standardized within- or between-schools. For example, it is unclear if institutions have discretion in how they formulate their institutional MCAT average, including how individuals with multiple test results are handled, which can introduce bias into the relationship between MCAT and USMLE performance [26].

Although we assessed numerous student and medical school factors, some potentially important covariates–such as percent of students with advanced degrees, curricular structure, timing of USMLE examinations, and school age–were not incorporated into this study but are important areas for future investigation. For example, some institutions have moved the USMLE Step 1 test window to after core clinical clerkships [23], with small benefits in scores and reduced failure rates [27].

Moreover, only 89 of 131 US allopathic medical schools (68%) reported complete data; non-reporters may differ in important ways. We found that reporting institutions, as compared to an average of all students nationally, had slightly higher average GPA and MCAT scores, with an associated 1.5-point higher average USMLE Step 1 score. Statistical comparisons of these differences are not advisable given the different units of reporting (institutions vs. individuals); yet the very small differences suggest that the reporting institutions were nationally representative. The relatively short 3-year study period does not preclude that the observed institutional outliers may represent random variation; replication with longer observation is needed. Finally, our study was ecological; no inference can be made that institution-

level findings translate to individual students (i.e., the ecological fallacy), and indeed only institutional averages and counts were reported (without any measure of student-to-student variability). Nevertheless, the purpose of this study was only to compare institutions.

## Conclusions

We found that institutional average GPA and MCAT scores correlate strongly with institutional USMLE performance. Numerous student demographic and institutional factors were insignificant when controlling for GPA and MCAT. We identified several institutions with significant above- or below-predicted USMLE performance, raising questions about inter-institutional USMLE performance parity. Methods to assess institutions' overall performance on knowledge-based exams may offer a parameter to evaluate medical schools and their curricula, while providing prospective students with valuable data regarding these high-stakes exams. Additional study is needed to explore the etiology and durability of the observed performance differences, and to incorporate other student and institutional factors that may be important predictors of performance.

## Supporting information

**S1 Table. Average GPA, MCAT, and USMLE Step 1 and 2 CK for 89 US allopathic medical schools reporting complete data from 2012–2014, with associated national averages.** (XLSX)

## Acknowledgments

This work was inspired by an anonymous post on March 13, 2013, in the Anastomosed blog titled: "Value added in top medical schools? MCAT/GPA as predictors of USMLE scores" (see https://anastomosed.wordpress.com/). Joel Purkiss, PhD (Baylor College of Medicine, Houston, Texas, USA) provided early guidance on the design of this work. Martin V. Pusic, MD, MA (New York University Langone Health, New York, USA) and Kent Hecker, PhD (University of Calgary, Alberta, Canada) provided helpful methodologic advice and manuscript critiques.

## Author Contributions

**Conceptualization:** Jesse Burk-Rafel, Joseph C. Kolars.

**Data curation:** Jesse Burk-Rafel.

**Formal analysis:** Jesse Burk-Rafel, Ricardo W. Pulido, Yousef Elfanagely.

**Investigation:** Jesse Burk-Rafel.

**Methodology:** Jesse Burk-Rafel, Joseph C. Kolars.

**Supervision:** Joseph C. Kolars.

**Validation:** Jesse Burk-Rafel.

**Visualization:** Jesse Burk-Rafel, Ricardo W. Pulido, Yousef Elfanagely.

**Writing – original draft:** Jesse Burk-Rafel, Ricardo W. Pulido, Yousef Elfanagely.

**Writing – review & editing:** Jesse Burk-Rafel, Ricardo W. Pulido, Yousef Elfanagely, Joseph C. Kolars.

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
