## [Decision Letter · Decision Letter 0]

5 Sep 2019

PONE-D-19-20613

Institutional differences in USMLE performance: Cross-sectional study of 89 US allopathic medical schools

PLOS ONE

Dear Dr. Burk-Rafel,

Thank you for submitting your manuscript to PLOS ONE. After careful consideration, we feel that it has merit but does not fully meet PLOS ONE’s publication criteria as it currently stands. Therefore, we invite you to submit a revised version of the manuscript that addresses the points raised during the review process.

We would appreciate receiving your revised manuscript by Oct 20 2019 11:59PM. To enhance the reproducibility of your results, we recommend that if applicable you deposit your laboratory protocols in protocols.io, where a protocol can be assigned its own identifier (DOI) such that it can be cited independently in the future. For instructions see: http://journals.plos.org/plosone/s/submission-guidelines#loc-laboratory-protocols

We look forward to receiving your revised manuscript.

Kind regards,

Andrew Carl Miller

Academic Editor

PLOS ONE

Journal Requirements:

"I have read the journal's policy and the authors of this manuscript have the following competing interests: Dr. Burk-Rafel reports working as a research consultant for ScholarRx, a digital learning platform that includes USMLE preparation services, during the late stages of writing this manuscript. ScholarRx was not involved in this study in any way. All other authors declare no competing interests."

Additional Editor Comments:

Please ensure that your manuscript adheres to the STROBE guidelines and provide citation:

von Elm E, Altman DG, Egger M, Pocock SJ, Gotzsche PC, Vandenbroucke JP. The Strengthening the Reporting of Observational Studies in Epidemiology (STROBE) Statement: guidelines for reporting observational studies. PLoS Med. 2007;4(10):e296. PMID: 17941714

Please indicate the study’s design with a commonly used term in the title or the abstract (eg. cohort, observational, retrospective, etc.)

Was a sample size calculation performed? If so, please provide.

Is the same data available for osteopathic medical schools? If so, why were they excluded?

42 allopathic schools were excluded from the analysis for not providing complete USMLE schools over the 3-year period. It would be nice to know more about those schools. Are there any commonalities amongst those schools that make them different from the other 89? Were they more likely to be new schools, struggling schools, etc. How do you think that this impacts the results?

How does school age impact the results (e.g. first 5 years vs longer established)?

Does the percentage of international students affect the results?

Do schools with a higher percentage of students with other advanced degrees (masters, PhD, etc) perform better (or worse)?

Is there a way to carry this forward to see the specialties that these students choose? Is there any relationship to schools having a higher or lower percentage of students matriculating into highly vs. less competitive specialties for residency?

Reviewers' comments:

Reviewer's Responses to Questions

**Comments to the Author**

1. Is the manuscript technically sound, and do the data support the conclusions?

Reviewer #1: Yes

Reviewer #2: Partly

2. Has the statistical analysis been performed appropriately and rigorously? 

Reviewer #1: I Don't Know

Reviewer #2: Yes

3. Have the authors made all data underlying the findings in their manuscript fully available?

Reviewer #1: No

Reviewer #2: Yes

4. Is the manuscript presented in an intelligible fashion and written in standard English?

Reviewer #1: Yes

Reviewer #2: Yes

5. Review Comments to the Author

Reviewer #1: It would be helpful to have a Table in the supporting document list the mean USMLE scores and mean MCAT scores for each institution since those data were the data used for the analysis. All of these data are publicly available why not list them.

Reviewer #2: Thank you for the opportunity to review this manuscript. This addresses an important topic in continuing to explore the impact of medical school curricula and other factors that contribute to USMLE interinstitutional variation. This large scale study does have major limitations which deserve greater discussion. Specific comments are listed below.

1. Page 5: describe where data will be available.

2. Methods: Further information is needed to better understand the data limitations of the self-reported data. For example, is anything known about consistency of reporting of the MCAT score? Is this is superscore of the last 3 takes or single best take? Is there a consistent reporting structure between institutions or is this reported completely at the institution’s discretion? The variability in how this data was reported represents a major potential limitation of the study.

3. Undergraduate gpa is a frequently discussed variable for MCAT performance. However, an increasing number of students have graduate degrees. Does percentage of students with graduate degrees and graduate gpa warrant a discussion in this manuscript?

4. Please describe how the MCAT score data was defined. Was this related to a single section of the MCAT or an average of the 3 sections?

5. In the results section, for overall interpretation of the results of this study, it would be important to understand the national USMLE average scores for the time period investigated. Is this sample representative of all medical schools?

6. During the past couple of decades, most medical schools have undergone significant curricular changes related to increased emphasis on active learning and decreasing lecture time. Many medical schools and students feel that the emphasis on USMLE performance competes with a curriculum focused on clinical performance. In future studies, it will be important to explore if certain types of curricula are more or less successful in preparing students for USMLE examinations.

Thank you again for the opportunity to review this manuscript. Despite the limitations, I feel that this type of larger study is needed to better understand USMLE performance at the institutional level.

6. PLOS authors have the option to publish the peer review history of their article (what does this mean?). If published, this will include your full peer review and any attached files.

Reviewer #1: No

Reviewer #2: No

---

## [Author Response · Author response to Decision Letter 0]

16 Oct 2019

Response to Reviewers (see attached 'Response to Reviewers' document for formatted version)

RE: PONE-D-19-20613

“Institutional differences in USMLE Step 1 and 2 CK performance: Cross-sectional study of 89 US allopathic medical schools”

In response to the reviewers’ comments, we addressed each point.

1. Please amend either the title on the online submission form (via Edit Submission) or the title in the manuscript so that they are identical.

The title of the online submission form and manuscript is “Institutional differences in USMLE Step 1 and 2 CK performance: Cross-sectional study of 89 US allopathic medical schools.” We changed the title on the online submission form and manuscript to be identical. 

2. Please ensure that your manuscript adheres to the STROBE guidelines and provide citation.

We have reviewed the STROBE guidelines and cited the citation into our manuscript. We attest that the manuscript adheres to the STROBE guidelines and we emphasize that our manuscript is purely an exploratory observational study meant to stimulate and generate a hypothesis based on reported data from prior years. The methodology of our study should still be generalizable to future studies as new data is reported. 

3. Please indicate the study’s design with a commonly used term in the title or the abstract (eg. cohort, observational, retrospective, etc.)

To further indicate the study’s design, we used the term “observational” in the abstract. The title already specifies it as “cross-sectional,” a type of observational study. 

4. Was a sample size calculation performed? If so, please provide.

A sample size calculation was not performed because we sought to obtain a complete census of all US allopathic medical schools. A statement to this effect was added to the methods.

5. Is the same data available for osteopathic medical schools (DO schools)? If so, why were they excluded?

Osteopathic institutions were excluded from this study, as osteopathic students historically took the COMLEX licensing examination rather than the USMLE and very few US osteopathic institutions reported USMLE performance data. It is important to note as well that at the time of data collection only 25 DO institutions had at least a first admitting class year, of these approximately 15 osteopathic schools were formed from 2002-2012. The rapid development and the continuous changes in curricula as typically seen in new institutions were additional reasons for excluding the small number of DO institutions that did report data. Theses reasons for excluding osteopathic institutions are now included in the methods.

6. 42 allopathic schools were excluded from the analysis for not providing complete USMLE schools over the 3-year period. It would be nice to know more about those schools. Are there any commonalities amongst those schools that make them different from the other 89? Were they more likely to be new schools, struggling schools, etc. How do you think that this impacts the results?

We have added national average performance data to Table 1 and S1 Table, and included a statement in results: “The average GPA, MCAT scores, and USMLE Step 1 scores for the 89 reporting institutions were slightly higher than national averages for all matriculants/examinees.” We go on in the discussion to discuss: “We found that reporting institutions, as compared to an average of all students nationally, had higher average GPA and MCAT scores, with an associated 1.5-point higher average USMLE Step 1 score. Statistical comparisons of these differences are not advisable given the different units of reporting (institutions vs. individuals); yet the very small differences suggest that the reporting institutions were nationally representative.” Overall, we are pleased that the reporting institutions very nearly matched national averages, suggesting good representativeness. Unfortunately, we do not have information regarding the non-reporting institutions to characterize their reasons for not reporting to US News and World Report.

7. How does school age impact the results (e.g. first 5 years vs longer established)?

We do not have this data. We have added this to the discussion as an important limitation and area for further investigation.

8. Does the percentage of international students affect the results?

Unfortunately, schools did not report the percent of international students within their classes. This would be an interesting covariate for future study. 

9. Do schools with a higher percentage of students with other advanced degrees (masters, PhD, etc) perform better (or worse)?

Schools' percent of non-traditional students (who took >1 year to do something between undergraduate and medical school) was not associated with differential performance, when controlling for MCAT and GPA. We do not have data on the percent of advanced degrees when entering medical school, but have added this as a limitation and area for future investigation. 

10. Is there a way to carry this forward to see the specialties that these students choose? Is there any relationship to schools having a higher or lower percentage of students matriculating into highly vs. less competitive specialties for residency?

Unfortunately, the main residency match data does not include match information at the level of schools and schools did not report distribution of specialties at match in this data set. However, the data set did include the percent of graduates entering primary care specialties (here defined as family medicine, pediatrics, and internal medicine), which have lower USMLE screening thresholds and are thus less “competitive”. Although having more primary care grads had a significant negative correlation with USMLE scores at baseline, it was not significant after controlling for GPA and MCAT. We have included several sentences in the discussion on this topic.

11. It would be helpful to have a Table in the supporting document list the mean USMLE scores and mean MCAT scores for each institution since those data were the data used for the analysis. All of these data are publicly available why not list them.

We have created a supporting table which lists the mean GPA, MCAT, and USMLE scores for all 89 institutions. This table is referred to in the results portion of our paper and included in the Supporting Information. 

12. Describe where data will be available.

Data will be available in the Supporting Information table.

13. Further information is needed to better understand the data limitations of the self-reported data. For example, is anything known about consistency of reporting of the MCAT score? Is this a superscore of the last 3 takes or single best take? Is there a consistent reporting structure between institutions or is this reported completely at the institution’s discretion? The variability in how this data was reported represents a major potential limitation of the study.

We do not know how schools reported their MCAT scores. We have added this to the limitations’ discussion of the paper and understand its importance. In a study conducted by Zhao et al, four different approaches were investigated when attempting to predict USMLE Step 1 scores using MCAT scores. It was concluded that averaging MCAT scores for an individual is the best predictor for Step 1 scores, highlighting the importance of how MCAT scores are gathered. However, it is reassuring that the average scores (GPA, MCAT, and USMLE) for all reporting institutions very nearly matched national averages for the study period, suggesting good fidelity of reporting.

14. Undergraduate GPA is a frequently discussed variable for MCAT performance. However, an increasing number of students have graduate degrees. Does the percentage of students with graduate degrees and graduate gpa warrant a discussion in this manuscript?

We do not have data concerning the percentage of students with graduate degrees and graduate GPA. We did note that the schools' percent of non-traditional students (who took >1 year to do something between undergraduate and medical school) was not associated with differential performance, when controlling for MCAT and GPA. We do not have data on the percent of advanced degrees when entering medical school, but added this as an important factor for future research. A literature review was performed to further investigate students with graduate degrees and their graduate GPA. No papers were found discussing the relationship between graduate GPA and medical school examinations, highlighting this as an important area for further investigation. 

15. Please describe how the MCAT score data was defined. Was this related to a single section of the MCAT or an average of the 3 sections?

The MCAT score data was defined as the average institutional scores for each of the 3 sections (biological sciences, physical sciences, verbal reasoning) to create a total score institutional average. We included this description of the MCAT score data to the methods. 

16. In the results section, for overall interpretation of the results of this study, it would be important to understand the national USMLE average scores for the time period investigated. Is this sample representative of all medical schools?

We have included the national average GPA, MCAT, and USMLE scores for the study period in Table 1 and S1 Table for reference. We have also added reflection in the results and limitations section regarding the representativeness of the sample, highlighting that the sample had very similar scores compared to national averages, suggesting a nationally representative sample. 

17. During the past couple of decades, most medical schools have undergone significant curricular changes related to increased emphasis on active learning and decreasing lecture time. Many medical schools and students feel that the emphasis on USMLE performance competes with a curriculum focused on clinical performance. In future studies, it will be important to explore if certain types of curricula are more or less successful in preparing students for USMLE examinations.

We are aware that some medical schools have undergone significant curricular changes. More specifically, there has been a trend in increasing clinical experience and decreasing time in the classroom. Studies completed by Daniel et al and Jurich et al have discussed the pros and cons experienced with this fundamental change in curriculum. We have added this information to the discussion portion of our study and recommended future studies explore which types of curricula are more or less successful in preparing students for USMLE examinations. 

18. The one suggestion is to clearly state in the conclusion that institutional mean GPA and mean MCAT scores correlate strongly with USMLE performance whereas student body demographic and other institutional variables were weak or insignificant when controlling for MCAT and GPA.

We updated the conclusion to clearly state that the institutional mean GPA and mean MCAT scores correlate strongly with USMLE performance whereas student body demographic and other institutional variables were weak or insignificant when controlling for MCAT and GPA.

---

## [Editor Report · Decision Letter 1]

21 Oct 2019

Institutional differences in USMLE Step 1 and 2 CK performance: Cross-sectional study of 89 US allopathic medical schools

PONE-D-19-20613R1

Dear Dr. Burk-Rafel,

We are pleased to inform you that your manuscript has been judged scientifically suitable for publication and will be formally accepted for publication once it complies with all outstanding technical requirements.

With kind regards,

Andrew Carl Miller

Academic Editor

PLOS ONE

Additional Editor Comments (optional):

Thank you for the important and interesting article.
---

## [Editor Report · Acceptance letter]

23 Oct 2019

PONE-D-19-20613R1 

Institutional differences in USMLE Step 1 and 2 CK performance: Cross-sectional study of 89 US allopathic medical schools 

Dear Dr. Burk-Rafel:

I am pleased to inform you that your manuscript has been deemed suitable for publication in PLOS ONE. Congratulations! Your manuscript is now with our production department. 

With kind regards,

on behalf of

Dr. Andrew Carl Miller 

Academic Editor

PLOS ONE